# Investigating the inflammation marker neutrophil-to-lymphocyte ratio in Danish blood donors with restless legs syndrome

**Joseph Dowsett**[1]*, **Maria Didriksen**[1], **Margit Hørup Larsen**[1], **Khoa Manh Dinh**[2], **Kathrine Agergård Kaspersen**[2,3], **Susan Mikkelsen**[2], **Lise Wegner Thørner**[1], **Erik Sørensen**[1], **Christian Erikstrup**[2], **Ole Birger Pedersen**[4], **Jesper Eugen-Olsen**[5], **Karina Banasik**[6], **Sisse Rye Ostrowski**[1]

1 Department of Clinical Immunology, Copenhagen University Hospital, Rigshospitalet, Copenhagen, Denmark, 2 Department of Clinical Immunology, Aarhus University Hospital, Aarhus, Denmark, 3 Danish Big Data Centre for Environment and Health (BERTHA), Aarhus University, Roskilde, Denmark, 4 Department of Immunology, Zealand University Hospital, Køge, Denmark, 5 Department of Clinical Research, Copenhagen University Hospital Amager and Hvidovre, Hvidovre, Denmark, 6 Novo Nordisk Foundation Center for Protein Research, Faculty of Health and Medical Sciences, University of Copenhagen, Copenhagen, Denmark

* joseph.dowsett@regionh.dk

**Data Availability Statement:** Data cannot be shared publicly as approval for public data-sharing were not obtained from participants. Data is

## Abstract

### Background

Restless Legs Syndrome (RLS) is a neurological sensorimotor disorder that occurs in the evening and night, thereby impacting quality of sleep in sufferers. The pathophysiology of RLS is poorly understood but inflammation has been proposed as possibly being involved. The neutrophil-to-lymphocyte ratio (NLR) can be used as an inflammation marker but results from small studies have been inconclusive in determining whether NLR is associated with RLS. We aimed to assess whether an association between NLR and RLS exists in a large cohort of healthy individuals.

### Methods

Neutrophils and lymphocytes were measured in blood samples of 13,055 individuals from the Danish Blood Donor Study, all of whom completed the validated Cambridge-Hopkins RLS-questionnaire for RLS assessment.

### Results

In the sample, 661 individuals were determined as current RLS cases (5.1%). A higher proportion of individuals with RLS were females (62.5% vs 47.5%; P<0.001) and RLS cases were older than controls (P<0.001), but no differences in body mass index (BMI), smoking or alcohol consumption were found between the two groups. An increased NLR was observed in RLS cases compared to controls (median NLR: 1.80 vs 1.72; P = 0.033). In an unadjusted logistic regression model, increased NLR was associated with RLS (OR = 1.10 per NLR unit increase [95%CI:1.01–1.20]; P = 0.032); however, the association was not

available for qualified scientists; for information on further access to data, please contact Professor Sisse Rye Ostrowski (from the Danish Blood Donor Study coordination committee) for data from the Danish Blood Donor Study (Sisse.Rye. Ostrowski@regionh.dk), or at the general DBDS contact email address: info@dbds.dk.

**Funding:** This study was supported by: -the Danish Council for Independent Research (09-069412 and 0602-02634B), https://dff.dk/en -the Bio- and Genome Bank Denmark https://www.regioner.dk/rbgben -the Novo Nordisk Foundation (NNF17OC0027594) https://novonordiskfonden.dk/en/ - K.B. acknowledges the Novo Nordisk Foundation (NNF14CC0001). https://novonordiskfonden.dk/en/ The funders had no role in study design, data collection and analysis, decision to publish, or preparation of the manuscript.

**Competing interests:** The authors have declared that no competing interests exist.

significant in multivariate models adjusting for sex and age (P = 0.094) or sex, age, alcohol consumption, smoking status and BMI (P = 0.107).

## Conclusion

We found no association between RLS and NLR among Danish blood donors after adjusting for sex, age, alcohol consumption, smoking status and BMI. Further studies are needed to determine whether inflammation is a risk factor for RLS.

## Introduction

The neurological sensorimotor disorder Restless Legs Syndrome (RLS) causes a persistent and irresistible urge to move one's legs that occurs or worsens in the evening and night, thereby often impacting quality of sleep in sufferers [1]. While the disorder is common, with a prevalence between 5 to 19% in European populations [2–5], options for treatment are currently limited and address symptoms rather than the underlying cause of RLS [6, 7]. The pathophysiology of RLS is poorly understood but increased understanding of it may inspire new and more effective treatment strategies. Current accepted pathways involved in the development of RLS include genetic predisposition, iron dysregulation in the central nervous system, and dopaminergic dysfunction [8, 9]. Inflammation has also been proposed as being involved in the pathophysiology of RLS as many conditions and diseases that are highly associated with RLS also have links to inflammation [10]. However, to date, only a limited number of studies have investigated links between inflammation and RLS. The neutrophil-to-lymphocyte ratio (NLR) can be used as an inflammation marker but results from small studies have been inconclusive in determining whether NLR is associated with RLS. Varım, Acar [11] found in a study of 75 RLS cases and 56 controls that NLR was higher in RLS cases than in controls, and suggested that NLR may be used as a predictor of RLS. On the contrary, Tak and Sengul [12] found no difference in NLR between RLS cases and controls in a similarly small-sized study (62 RLS cases, 40 controls). Due to these discrepant findings in small studies, we aimed to assess whether an association between NLR and RLS exists in a large cohort of 13,055 otherwise healthy individuals, including 661 RLS cases. As RLS is known to impact quality of sleep, and sleep disturbances have been shown to increase systemic inflammation markers [13], we also investigated whether NLRs are increased in donors reporting having difficulty sleeping. We also evaluated the associations with RLS for nine other haematological measurements, including white blood cell (WBC) count, platelets and platelet-to-lymphocyte ratio (PLR).

## Methods

### Study population

Individuals that participated in this study were part of the Danish Blood Donor Study (DBDS), a nationwide research platform utilizing the existing infrastructure in the Danish blood banks by including blood donors when they show up to donate [14]. Participants were between the ages of 18 and 67 years and were required to be generally healthy to be eligible as blood donors. Blood donors are permanently excluded from blood donation if diagnosed with chronic diseases such as diabetes, cancer, cardiovascular diseases including hypertension and statin-treated hypercholesterolemia, autoimmune diseases, hepatitis, chronic respiratory diseases, kidney diseases and metabolic diseases. At every donation, the blood donors are asked whether

they have consulted a doctor since last donation. Upon enrolment, participants gave written informed consent, whole blood, plasma, and answered a comprehensive questionnaire, including lifestyle factors such as smoking habits, alcohol consumption, height and weight (for BMI). The project was approved by the Research Ethics Committee (1-10-72-95-13) in Central Denmark Region and by the Danish Data Protection Agency under the combined approval for healthcare research at The Capital Region of Denmark (P-2019-99).

### Questionnaire data

RLS-status was determined using the Cambridge-Hopkins RLS-questionnaire (CH-RLSq) containing 10 items and has been validated in several population settings [15]. The questionnaire was translated from English to Danish using the back-translation method as previously described [5]. A total of 52,921 DBDS participants answered the CH-RLSq. The donor must have experienced the RLS symptoms within the past 12 months to be considered a current RLS case. Participants with correctly completed CH-RLSq, complete covariate data, and haematological measurements available were included in the analysis (see S1 Fig for flowchart).

In addition to the questions on the CH-RLSq, the donors were also asked to complete some additional questionnaires. As part of the Major Depression Inventory MDI questionnaire [16], donors were asked the question "Over the last two weeks, how much of the time have you had trouble sleeping at night?". Donors were given the following six options as their answer: "at no time", "some of the time", "less than half of the time", "more than half of the time", "most of the time" and "all of the time". We converted the variable into a binary variable, as specified previously [16]. A donor was classified as having difficulty sleeping when they answer either "more than half of the time", "most of the time" or "all of the time" to the question.

### Haematological measurements

Haematological values were measured in blood samples from the same donation visit as the CH-RLSq was completed using a commercially available assay (Sysmex XT-1800i, Kobe, Japan). The absolute neutrophil and lymphocyte count were measured in 13,055 participants from the Central Denmark Region (Region Midtjylland). The NLR was calculated by dividing the absolute neutrophil count by the absolute lymphocyte count. A NLR below 3.5 is considered normal in an adult, non-geriatric population [17] and therefore we also created a binary NLR variable using a cut-off of 3.5. For the supplementary analyses, additional haematological measurements from the Sysmex assay were available (N = 28,871), originating from both the Central Denmark Region and the Capital Region of Denmark (S1 Fig). These measurements include absolute platelet count, platelet distribution width, platelet large cell ratio, absolute WBC count, mean corpuscular volume, mean corpuscular haemoglobin concentration and the PLR, calculated by dividing the absolute platelet count by the absolute lymphocyte count.

### Statistical analyses

Differences in demographic descriptive statistics and haematological measurements between RLS cases and controls were compared by calculating the median with interquartile ranges (IQR) for non-normally distributed quantitative variables, and count number with percentages for categorical variables. Comparisons between RLS cases and controls were made using the Chi-square test for categorical variables and Kruskal-Wallis rank test for continuous variables. For the analysis investigating sleeping difficulty and the NLR, comparisons were also made using the Kruskal-Wallis rank test. Logistic regression models were then employed to assess the effect of NLR and the other haematological values on the probability of having RLS. Three models were performed: Model 0 = Crude association; Model 1 = adjusting for sex and age;

Model 2 = adjustment for sex, age, alcohol consumption, smoking status and BMI. An association was considered statistically significant when the p-value was below 0.05. All statistical analyses were performed using R (version 4.0).

## Results

Neutrophil and lymphocyte data were available in 13,055 participants, of which 661 individuals were determined as current RLS cases (5.1%). RLS cases included a higher proportion of females (62.5% vs 47.5%; P<0.001) and were older than controls (median age 40.9 years in RLS cases vs 38.0 years in controls; P<0.001). No differences in BMI, smoking or alcohol consumption were found between RLS cases and controls (Table 1).

NLR was higher in RLS cases (median NLR: 1.80) compared to controls (median NLR: 1.72) (P = 0.033). Furthermore, RLS cases had higher absolute number of neutrophils (median neutrophil count: $3.51 \times 10^9$/L in RLS vs $3.40 \times 10^9$/L in controls; P = 0.007). No differences between the two groups were observed for absolute lymphocyte count or in the proportion of individuals with NLR above 3.5 (Table 1). To ensure that the increase in the NLR in RLS cases is not be due to secondary insomnia, we examined whether donors with self-reported difficulties sleeping had higher NLRs (Table 2 and S2 Fig). Of the 13,055 participants with NLR and RLS data available, only 39 (0.3%) did not answer the question on sleeping difficulty. We found that there was no significant difference (P = 0.772) in the NLR between donors having difficulty sleeping (NLR: 1.73 (IQR: 1.36–2.21)) and donors who do not (NLR: 1.73 (1.35–2.19)). In regard to RLS, we observed a significantly higher proportion of RLS cases reporting having difficulty sleeping (10.6%) compared to controls (5.0%) (P<0.001) (S1 Table and S3 and S4 Figs).

We performed logistic regression models using neutrophils, lymphocytes and NLR as the predictor variable and RLS status as the outcome (Fig 1). We found that an increased NLR was associated with RLS in the unadjusted model (OR for RLS = 1.10 per NLR unit increase [95% CI:1.01–1.20]; P = 0.032); however, the association was not significant after adjusting for sex and age (OR = 1.08 [0.99–1.19]; P = 0.094) or sex, age, alcohol consumption, smoking status and BMI (OR = 1.08 [0.98–1.18]; P = 0.107). Likewise, the association between absolute neutrophil count and RLS was not significant after adjusting for sex and age (OR = 1.04 per $1 \times 10^9$/L increase [0.97:1.10]; P = 0.270).

We additionally analysed seven other haematological measurements in 28,871 participants (S2 Table). Compared to controls, RLS cases had higher mean corpuscular volumes (P = 0.002) and lower mean corpuscular haemoglobin concentrations (P<0.001). However, as with NLR, the associations were no longer significant after adjusting for sex and age (S5 Fig).

## Discussion

We aimed to assess the association between NLR and RLS in a large cohort of healthy Danish blood donors. Although a significant difference in NLR was observed between RLS cases and controls, the association was not significant in the logistic regression model when adjusting for sex and age or when adjusting for sex, age, alcohol, smoking and BMI.

NLR has previously been associated with RLS in a small study with 75 RLS cases and 56 controls [11]. However, a later study with 62 RLS cases and 40 controls could not replicate the association [12]. Our study is strengthened by our superior sample size of 13,055 individuals including 661 RLS cases, which provides strong evidence against any association between NLR and RLS after only correcting for sex and age. Importantly, none of the former studies adjusted for age and sex [11, 12]. Using a uniformly healthy study population is also a clear strength of our study, as blood donors are thoroughly screened at every visit to the blood bank. RLS can

**Table 1. Demographic descriptive statistics for RLS cases and controls in the Danish Blood Donor Study with neutrophil, lymphocyte and NLR data available (N = 13,055).**

|  | Controls | | RLS Cases | | P value[a] |
|---|---|---|---|---|---|
|  | N = 12,394 | | N = 661 (5.1%) | |  |
|  | N | % | N | % |  |
| **Demographic characteristics** | | | | | |
| **Sex** | | | | | |
| Male | 6,508 | 52.5 | 248 | 37.5 | <0.001 |
| Female | 5,886 | 47.5 | 413 | 62.5 | |
| **Age** | | | | | |
| median (IQR) | 38.0 (27.2–49.9) | | 40.9 (29.1–51.5) | | <0.001 |
| **BMI** | | | | | |
| median (IQR) | 24.8 (22.6-27-4) | | 24.9 (22.7–28.0) | | 0.104 |
| <18.5 | 76 | 0.6 | 5 | 0.8 | 0.257 |
| 18.5–25 | 6,510 | 52.5 | 333 | 50.4 | |
| 25–30 | 4,365 | 35.2 | 225 | 34.0 | |
| 30–35 | 1,082 | 8.7 | 75 | 11.3 | |
| 35–40 | 281 | 2.3 | 18 | 2.7 | |
| >40 | 80 | 0.6 | 5 | 0.8 | |
| **Smoking status** | | | | | |
| Non-smoker | 10,961 | 88.4 | 573 | 86.7 | 0.268 |
| <1 cigarette per day | 518 | 4.2 | 28 | 4.2 | |
| >1 cigarette per day | 915 | 7.4 | 60 | 9.1 | |
| **Alcohol consumption** | | | | | |
| Never/almost never | 1,629 | 13.1 | 87 | 13.2 | 0.792 |
| A couple of times a month | 6,711 | 54.1 | 351 | 53.1 | |
| A couple of times a week | 3,617 | 29.2 | 195 | 29.5 | |
| Daily/almost daily | 437 | 3.5 | 28 | 4.2 | |
| **Haematological Measurements** | | | | | |
| Absolute neutrophil count $10^9$/L, median (IQR) | 3.40 (2.72–4.24) | | 3.51 (2.86–4.41) | | **0.007** |
| Absolute lymphocyte count $10^9$/L, median (IQR) | 1.96 (1.63–2.35) | | 2.00 (1.64–2.39) | | 0.427 |
| Neutrophil-to-Lymphocyte Ratio (NLR) Median (IQR) | 1.72 (1.35–2.21) | | 1.80 (1.39–2.25) | | **0.033** |
| NLR > 3.5 (% above cut-off) | 457 | 3.7 | 33 | 5.0 | 0.085 |

[a]For comparison of the two groups, chi-square test was used for categorical variables and Kruskal-Wallis rank test was used for continuous variables.

be categorised as either primary/idiopathic or secondary, in which secondary RLS has an identifiable cause such as kidney disease or pregnancy [18]. We therefore assume that RLS cases in the DBDS are predominantly idiopathic as eligible donors must be healthy which in turn reduces the presence of comorbidities associated with secondary RLS. Distinguishing between primary/idiopathic and secondary RLS may therefore explain why associations have previously been observed between inflammatory disorders and RLS [10]. However, it must be noted that CH-RLSq was used to determine RLS status in contrast to the International Restless Legs Syndrome Study Group (IRLSSG) questionnaire used in the previous studies. Whether our alternative method for RLS assessment impacted our results is unknown, though the CH-RLSq is also based on the IRLSSG criteria, and may also have additional strengths as it includes additional items and has been validated in several population settings (diagnostic sensitivity 87.2% and specificity 94.0%) [15].

**Table 2. Median NLR in blood donors reporting difficulty sleeping in the DBDS NLR-RLS dataset, excluding 39 who did not answer the question (N = 13,016).**

| | N | Neutrophil-to-Lymphocyte Ratio (NLR) | P value[a] |
|---|---|---|---|
| | | Median (IQR) | |
| **In the 2 weeks prior to donation, donors reported having difficulty sleeping. . .** | | | |
| A1. at no time | 6,826 | 1.72 (1.36–2.20) | 0.768 |
| A2. some of the time | 4,854 | 1.73 (1.35–2.21) | |
| A3. less than half of the time | 645 | 1.77 (1.35–2.23) | |
| A4. more than half of the time | 391 | 1.76 (1.40–2.22) | |
| A5. most of the time | 246 | 1.68 (1.29–2.15) | |
| A6. All of the time | 54 | 1.63 (1.33–2.15) | |
| **Difficulty Sleeping (Binary variable)** | | | |
| No (A1-A3) | 12,325 | 1.73 (1.36–2.21) | 0.772 |
| Yes (A4-A6) | 691 | 1.73 (1.35–2.19) | |

[a] Kruskal-Wallis rank test.

Our study also found no difference in the NLR between donors having difficulty sleeping and those who do not. Although sleep disturbances are associated with increased inflammation, a large meta-analysis on sleep disturbance, sleep duration, and inflammation in adults concluded that short sleep duration is not associated with increases in systemic inflammation markers [13], which is in line with our findings.

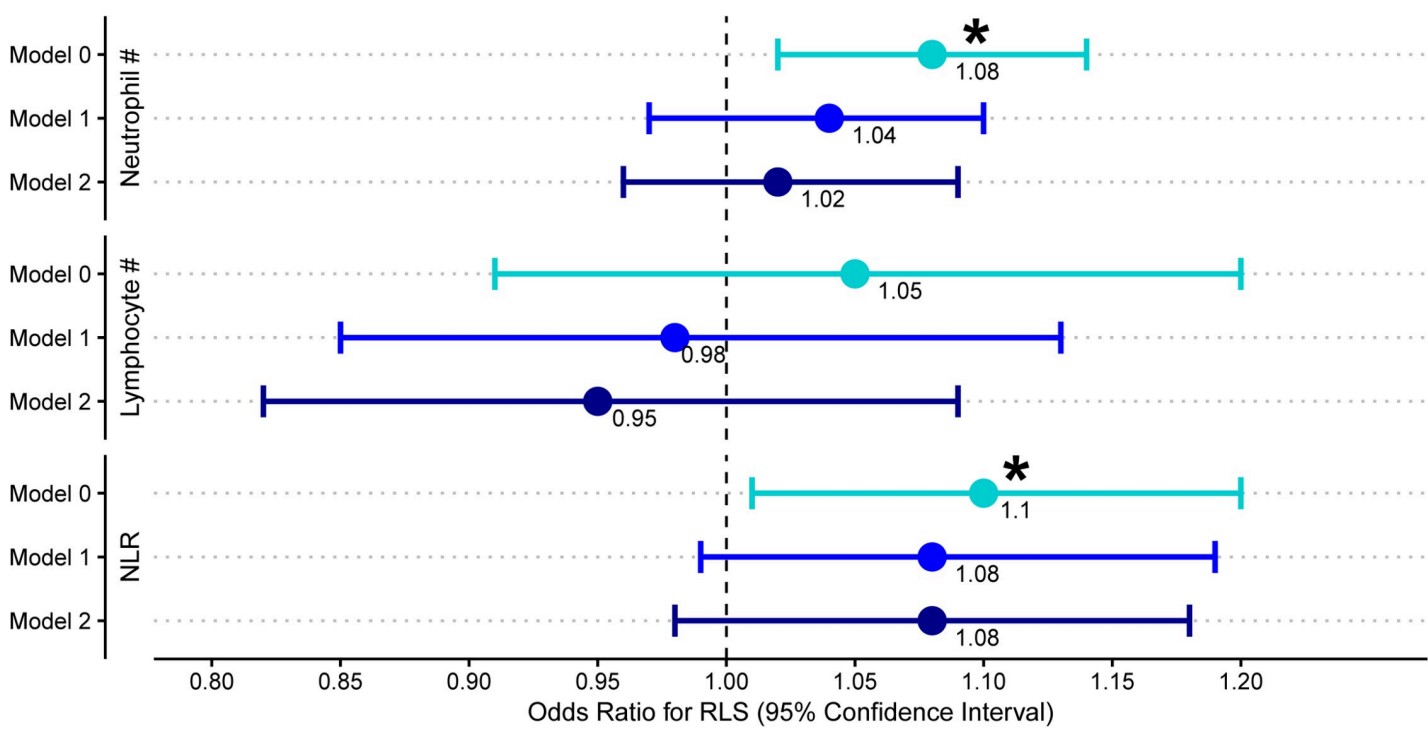

**Fig 1. Investigating the NLR in RLS.** Results from the logistic regression models using neutrophils, lymphocytes and NLR as the predictor variable and RLS status as the outcome in the DBDS cohort. Model 0 = Crude association. Model 1 = adjusting for sex and age. Model 2 = adjusting for sex, age, alcohol consumption, smoking and BMI. Data are presented as odd ratios (OR) with 95% confidence intervals and an asterisk (*) denotes a P value < 0.05.

Despite our findings that NLR was not associated with RLS after correcting for sex and age, a link between primary/idiopathic RLS and inflammation may still exist and may therefore be discoverable using other inflammatory markers than NLR. Associations between RLS and circulating levels of soluble inflammatory markers have been found previously in small studies, including C-reactive protein (CRP) [19, 20], interleukin-6 (IL-6) [21], interleukin-1beta [21] and tumour-necrosis factor alpha [21]. However, some studies have not been able to replicate the associations with CRP [21, 22] and IL-6 [22], emphasizing the need for further studies with larger samples sizes investigating inflammatory markers in RLS.

In conclusion, we found no association between RLS and the inflammatory marker NLR among Danish blood donors after adjusting for sex, age, alcohol consumption, smoking status and BMI, indicating that NLR has no role as a marker of RLS. Further studies exploring the potential link between RLS and inflammation investigating different inflammatory markers are warranted, to determine whether inflammation is involved in the pathophysiology of RLS.

## Supporting information

**S1 Fig. Flowchart of the inclusion process.**
(PDF)

**S2 Fig. Median NLR and difficulty sleeping.**
(PDF)

**S3 Fig. Proportion of RLS cases and controls reporting that they have had difficulty sleeping.**
(PDF)

**S4 Fig. Proportion of RLS cases and controls reporting that they have had difficulty sleeping (as binary variable).**
(PDF)

**S5 Fig. Haematological markers as predictors of RLS.**
(PDF)

**S1 Table. Proportion of RLS cases and controls reporting that they have had difficulty sleeping.**
(PDF)

**S2 Table. Descriptive statistics for RLS cases and controls for other haematological markers.**
(PDF)

## Acknowledgments

We thank the Danish blood donors for their valuable participation in the Danish Blood Donor Study as well as the staff at the blood centres making this study possible.

## Author Contributions

**Conceptualization:** Joseph Dowsett, Maria Didriksen, Sisse Rye Ostrowski.

**Data curation:** Maria Didriksen, Margit Hørup Larsen, Khoa Manh Dinh, Kathrine Agergård Kaspersen, Susan Mikkelsen, Lise Wegner Thørner, Erik Sørensen, Christian Erikstrup, Ole Birger Pedersen.

**Formal analysis:** Joseph Dowsett.

**Funding acquisition:** Maria Didriksen, Christian Erikstrup, Ole Birger Pedersen.

**Investigation:** Joseph Dowsett.

**Methodology:** Joseph Dowsett, Margit Hørup Larsen, Khoa Manh Dinh, Kathrine Agergård Kaspersen, Susan Mikkelsen, Erik Sørensen.

**Project administration:** Lise Wegner Thørner, Erik Sørensen, Christian Erikstrup, Ole Birger Pedersen, Sisse Rye Ostrowski.

**Supervision:** Jesper Eugen-Olsen, Karina Banasik, Sisse Rye Ostrowski.

**Visualization:** Joseph Dowsett.

**Writing – original draft:** Joseph Dowsett.

**Writing – review & editing:** Joseph Dowsett, Maria Didriksen, Margit Hørup Larsen, Khoa Manh Dinh, Kathrine Agergård Kaspersen, Susan Mikkelsen, Lise Wegner Thørner, Erik Sørensen, Christian Erikstrup, Ole Birger Pedersen, Jesper Eugen-Olsen, Karina Banasik, Sisse Rye Ostrowski.

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
