## [Decision Letter · Decision Letter 0]

18 Oct 2021

PONE-D-21-27529Investigating the inflammation marker neutrophil-to-lymphocyte ratio in Danish blood donors with restless legs syndromePLOS ONE

Dear Dr. Dowsett,

Thank you for submitting your manuscript to PLOS ONE. After careful consideration, we feel that it has merit but does not fully meet PLOS ONE’s publication criteria as it currently stands. Therefore, we invite you to submit a revised version of the manuscript that addresses the points raised during the review process.

 Specifically it was thought that inclusion of information regarding sleep quality would be informative, and so addition of data or at least a discussion of this concept is recommended.

We look forward to receiving your revised manuscript.

Kind regards,

Colin Johnson, Ph.D.

Academic Editor

PLOS ONE

Journal Requirements:

a) Did participants provide their written or verbal informed consent to participate in this study?

Reviewers' comments:

Reviewer's Responses to Questions

**Comments to the Author**

1. Is the manuscript technically sound, and do the data support the conclusions?

Reviewer #1: Yes

2. Has the statistical analysis been performed appropriately and rigorously? 

Reviewer #1: Yes

3. Have the authors made all data underlying the findings in their manuscript fully available?

Reviewer #1: Yes

4. Is the manuscript presented in an intelligible fashion and written in standard English?

Reviewer #1: Yes

5. Review Comments to the Author

Reviewer #1: I have read the manuscript with great interests. I believe that Authors' findings deserve publication.

I have following remarks:

1. There is no data about quality of sleep and prevalence if insomnia in the studied population. RLS is related to shortened sleep. Shortened sleep is related to activity of immunological system. Therefore I would suggest to show data about sleep quality in the examined population and to include them in statistical analysis (e.g. higher NLR may result from secondary insomnia) If such data is not available I would suggest commenting that issue in Discussion

2. The Authors write that it was "healthy population of blood donors". Where do we know it from? Can we be sure that Danish blood donors do not have any chronic conditions that may be related to chronic inflammation?

3. I suggest disputing this finding in context of prevalence of RLS in chronic neurologic or systematic inflammatory disorders

6. PLOS authors have the option to publish the peer review history of their article (what does this mean?). If published, this will include your full peer review and any attached files.

Reviewer #1: **Yes: **Mariusz Sieminski

---

## [Author Response · Author response to Decision Letter 0]

20 Oct 2021

Please see the Rebuttal letter document. The responses have been copy/pasted below:

Reviewer 1:

I have read the manuscript with great interests. I believe that Authors' findings deserve publication.

RESPONSE: We thank the reviewer for their comments and interest in our manuscript’s findings. We are grateful for their constructive comments and we believe the reviewer’s three minor points have helped to improve our manuscript.

1. There is no data about quality of sleep and prevalence if insomnia in the studied population. RLS is related to shortened sleep. Shortened sleep is related to activity of immunological system. Therefore I would suggest to show data about sleep quality in the examined population and to include them in statistical analysis (e.g. higher NLR may result from secondary insomnia) If such data is not available I would suggest commenting that issue in Discussion.

RESPONSE: We agree with the reviewer that showing sleep quality data in the examined population would be informative. We have therefore investigated whether NLR is associated with reported sleep quality in our cohort. 

We have now added these new analyses to the manuscript.

On page 2, lines 55-57 in the introduction:

“As RLS is known to impact quality of sleep, and sleep disturbances have been shown to increase systemic inflammation markers(13), we also investigated whether NLRs are increased in donors reporting having difficulty sleeping.”.

On page 4, lines 85-92 in the methods:

“In addition to the questions on the CH-RLSq, the donors were also asked to complete some additional questionnaires. As part of the Major Depression Inventory MDI questionnaire (16), donros were asked the question “Over the last two weeks, how much of the time have you had trouble sleeping at night?”. Donors were given the following six options as their answer: “at no time”, “some of the time”, “less than half of the time”, “more than half of the time”, “most of the time” and “all of the time”. We converted the variable into a binary variable, as specified previously(16). A donor was classified as having difficulty sleeping when they answer either “more than half of the time”, “most of the time” or “all of the time” to the question.”

On page 5, lines 112-114 in the methods:

“For the analysis investigating sleeping difficulty and the NLR, comparisons were also made using the Kruskal-Wallis rank test.”

On page 7, lines 135-142 in Results:

“To ensure that the increase in the NLR in RLS cases is not be due to secondary insomnia, we examined whether donors with self-reported difficulties sleeping had higher NLRs (Table 2 and Supp. Fig. 2). Of the 13,055 participants with NLR and RLS data available, only 39 (0.3%) did not answer the question on sleeping difficulty. We found that there was no significant difference (P=0.772) in the NLR between donors having difficulty sleeping (NLR: 1.73 (IQR: 1.36–2.21)) and donors who do not (NLR: 1.73 (1.35–2.19)). In regard to RLS, we observed a significantly higher proportion of RLS cases reporting having difficulty sleeping (10.6%) compared to controls (5.0%) (P<0.001) (Supp. Table 1 and Supp. Fig.3-4).”

On page 7, a new table (Table 2) has been added, with data on the median NLR and sleep quality. S2 Fig visualises the median NLR in each answer group, and we also looked at sleep quality in regard to RLS status (as expected, RLS cases have more difficulty sleeping) and showed these in the following table and figures: S1 Table, S3 and S4 Figs.

On page 10, lines 193-197 in Discussion:

“Our study also found no difference in the NLR between donors having difficulty sleeping and those who do not. Although sleep disturbances are associated with increased inflammation, a large meta-analysis on sleep disturbance, sleep duration, and inflammation in adults concluded that short sleep duration is not associated with increases in systemic inflammation markers(13), which is in line with our findings.”

2. The Authors write that it was "healthy population of blood donors". Where do we know it from? Can we be sure that Danish blood donors do not have any chronic conditions that may be related to chronic inflammation?

RESPONSE: We apologise for not making this clear. Danish blood donors must comply with strict criteria to be allowed to donate and are permanently excluded from blood donation if diagnosed with chronic diseases such as diabetes, cancer, hypertension, statin-treated hypercholesterolemia, cardiovascular disease, autoimmune diseases, hepatitis, chronic respiratory diseases, kidney diseases and metabolic diseases. 

Blood donors are only allowed to donate if they report that they feel perfectly healthy and they are to report use of medication. At every donation, the blood donors are asked again and whether they have consulted a doctor since last donation. We are therefore confident that the blood donors are healthy and that they do not have any chronic diseases that may be related to chronic inflammation. 

However, Danish blood donors are allowed to have a high BMI, be smokers and consume alcohol, which may be considered conditions that are relate to chronic inflammation, and we therefore adjust our statistical analyses for these variables. 

We have now made this more clear in the manuscript:

On page 3, lines 66-70:

“Participants were between the ages of 18 and 67 years and were required to be generally healthy to be eligible as blood donors. 

Blood donors are permanently excluded from blood donation if diagnosed with chronic diseases such as diabetes, cancer, cardiovascular diseases including hypertension and statin-treated hypercholesterolemia, autoimmune diseases, hepatitis, chronic respiratory diseases, kidney diseases and metabolic diseases. At every donation, the blood donors are asked whether they have consulted a doctor since last donation”.

3. I suggest disputing this finding in context of prevalence of RLS in chronic neurologic or systematic inflammatory disorders.

RESPONSE: Thank you for your suggestion.

We have added the following in the discussion:

Page 9, lines 180-186:

“RLS can be categorised as either primary/idiopathic or secondary, in which secondary RLS has an identifiable cause such as kidney disease or pregnancy. We therefore assume that RLS cases in the DBDS are predominantly idiopathic as eligible donors must be healthy which in turn reduces the presence of comorbidities associated with secondary RLS. Distinguishing between primary/idiopathic and secondary RLS may therefore explain why associations have previously been observed between inflammatory disorders and RLS. ”

Page 10, lines 198-200 :

“Despite our findings that NLR was not associated with RLS after correcting for sex and age, a link between primary/idiopathic RLS and inflammation may still exist and may therefore be discoverable using other inflammatory markers than NLR.”

---

## [Decision Letter · Decision Letter 1]

25 Oct 2021

Investigating the inflammation marker neutrophil-to-lymphocyte ratio in Danish blood donors with restless legs syndrome

PONE-D-21-27529R1

Dear Dr. Dowsett,

We’re pleased to inform you that your manuscript has been judged scientifically suitable for publication and will be formally accepted for publication once it meets all outstanding technical requirements.

Kind regards,

Colin Johnson, Ph.D.

Academic Editor

PLOS ONE

Additional Editor Comments (optional):

Reviewers' comments:

Reviewer's Responses to Questions

**Comments to the Author**

1. If the authors have adequately addressed your comments raised in a previous round of review and you feel that this manuscript is now acceptable for publication, you may indicate that here to bypass the “Comments to the Author” section, enter your conflict of interest statement in the “Confidential to Editor” section, and submit your "Accept" recommendation.

Reviewer #1: All comments have been addressed

2. Is the manuscript technically sound, and do the data support the conclusions?

Reviewer #1: Yes

3. Has the statistical analysis been performed appropriately and rigorously? 

Reviewer #1: Yes

4. Have the authors made all data underlying the findings in their manuscript fully available?

Reviewer #1: Yes

5. Is the manuscript presented in an intelligible fashion and written in standard English?

Reviewer #1: Yes

6. Review Comments to the Author

Reviewer #1: I am satisfied with Authors' response. All my doubts were addressed. I find now this manucript eligible for publication.

7. PLOS authors have the option to publish the peer review history of their article (what does this mean?). If published, this will include your full peer review and any attached files.

Reviewer #1: **Yes: **Mariusz Siemiński

---

## [Editor Report · Acceptance letter]

3 Nov 2021

PONE-D-21-27529R1 

Investigating the inflammation marker neutrophil-to-lymphocyte ratio in Danish blood donors with restless legs syndrome 

Dear Dr. Dowsett:

I'm pleased to inform you that your manuscript has been deemed suitable for publication in PLOS ONE. Congratulations! Your manuscript is now with our production department. 

Kind regards, 

on behalf of

Dr. Colin Johnson 

Academic Editor

PLOS ONE